# One Size Does Not Fit All: The Past, Present and Future of Cystic Fibrosis Causal Therapies

**DOI:** 10.3390/cells11121868

**Published:** 2022-06-08

**Authors:** Marjolein M. Ensinck, Marianne S. Carlon

**Affiliations:** 1Molecular Virology and Gene Therapy, Department of Pharmaceutical and Pharmacological Sciences, KU Leuven, 3000 Leuven, Flanders, Belgium; marjolein.ensinck@kuleuven.be; 2Laboratory of Respiratory Diseases and Thoracic Surgery (BREATHE), Department of Chronic Diseases and Metabolism, KU Leuven, 3000 Leuven, Flanders, Belgium

**Keywords:** cystic fibrosis (CF), cystic fibrosis transmembrane conductance regulator (CFTR), personalized medicine, CFTR modulators, proteostasis modulation, stabilizers, amplifiers, translational readthrough inducing drugs (TRIDs), NMD inhibition, gene therapy

## Abstract

Cystic fibrosis (CF) is the most common monogenic disorder, caused by mutations in the CF transmembrane conductance regulator (*CFTR*) gene. Over the last 30 years, tremendous progress has been made in understanding the molecular basis of CF and the development of treatments that target the underlying defects in CF. Currently, a highly effective CFTR modulator treatment (Kalydeco™/Trikafta™) is available for 90% of people with CF. In this review, we will give an extensive overview of past and ongoing efforts in the development of therapies targeting the molecular defects in CF. We will discuss strategies targeting the CFTR protein (i.e., CFTR modulators such as correctors and potentiators), its cellular environment (i.e., proteostasis modulation, stabilization at the plasma membrane), the *CFTR* mRNA (i.e., amplifiers, nonsense mediated mRNA decay suppressors, translational readthrough inducing drugs) or the *CFTR* gene (gene therapies). Finally, we will focus on how these efforts can be applied to the 15% of people with CF for whom no causal therapy is available yet.

## 1. Introduction into Cystic Fibrosis

Cystic fibrosis (CF) is Europe’s most common life-threatening autosomal recessive disorder, which affects approximately 50,000 people in Europe and over 85,000 worldwide [1]. While CF in the past has mainly been a pediatric condition, the predicted mean survival of newborns with CF in 2019 was 48 years [2]. Morbidity and mortality in people with CF (PwCF) are mainly caused by progressive obstructive lung disease, with perpetual cycles of airway infection and inflammation, resulting in bronchiectasis, tissue remodeling and a decline in lung function over time [3,4]. Besides the airways, many other organs are affected in CF, including the exocrine pancreas, intestines and sweat glands (reviewed in [5]). Co-morbidities are becoming increasingly important in CF. They can either be congenital, such as congenital bilateral absence of vas deferens (CBAVD), or have a later onset due to progressive damage to organs, such as CF related diabetes (CFRD) and CF related liver disease (CFLD). CF is now, more than ever, a multi-organ disease, requiring therapeutic approaches that cover a wide range of organs.

CF symptoms are caused by an imbalance in ion and water homeostasis in the secretory epithelia of these organs due to loss of function of the apical chloride and bicarbonate channel CFTR (cystic fibrosis transmembrane conductance regulator, *see Section 2.1. CFTR—Structure, Folding and Function*). In the pancreas, the absence of CFTR-dependent fluid secretion leads to the destruction of the exocrine pancreatic glands at birth for 85% of PwCF [6]. In the airways, mucus is not readily removed due to a decrease in periciliary liquid height, a watery layer between cells and mucus that depends on CFTR chloride secretion and resulting osmosis and is essential for good mucociliary clearance. In addition, pH changes from lack of bicarbonate secretion prevent the proper unfolding of mucins, compacting the mucus and making it even harder to expel (reviewed in [7]). Combined with altered innate immune responses, such as loss of antimicrobial peptide function [8] and the overactive but ineffective response of infiltrating neutrophils, this creates an environment for opportunistic micro-organisms to thrive in ([9] and reviewed in [10]). Chances of chronic colonization with *Pseudomonas aeruginosa*, *Burkholderia cepacia*, *Staphylococcus aureus* and other common CF pathogens increase throughout the lifespan of PwCF, and by the age of 20, over 80% of PwCF are colonized with at least one micro-organism [2]. Chronic infection is correlated with a more rapid disease progression and worse prognosis [11], stressing the urgency of early treatments in PwCF, preferably before the onset of chronic infections.

In this review, we will discuss different strategies for the causal treatment of CF, both market-approved and in development. First, we will briefly discuss the CFTR protein and different mutations to stress the need of multiple strategies to cover all PwCF. Next, we will look into strategies to (1) improve CFTR function, (2) increase the amount of CFTR at the plasma membrane (PM), (3) increase the amount of immature CFTR protein or (4) provide a correct *CFTR* template. Finally, we will briefly discuss how to further personalize treatment for PwCF and bring causal therapies to the entire CF population.

## 2. One Size Does Not Fit All

A good understanding of the structure and function of the wildtype CFTR protein is essential for (1) grasping the molecular consequences of mutations that result in CF and (2) developing strategies using these models to correct the identified defects. As we will discuss in the next section, the multitude of mutations in *CFTR* identified and characterized to date will evidence the need of therapeutic strategies tailored to individual (classes of) molecular defects.

### 2.1. CFTR—Structure, Folding and Function

CFTR is a 1480 amino acid glycoprotein and member of the ATP-binding cassette (ABC) transporter family. Uniquely for ABC transporters, it functions as an ion channel rather than a transporter (reviewed in [12]). It is comprised of two halves containing a nucleotide binding domain (NBD) and a membrane spanning domain (MSD)—a common architecture for ABC transporters. The two halves, however, are linked via a regulatory (R) domain that is not found in other members of this protein family. 

Folding of the CFTR protein is a complex process which mainly takes place co-translationally, however inter-domain interactions are completed post-translationally ( [13,14] and reviewed in [15]). The endoplasmic reticulum (ER) quality control (ERQC) evaluates the folding-status of the nascent CFTR protein at several checkpoints [16]. Proteins with a non-native conformation are readily recognized by this machinery. They are primed via ubiquitination for ER associated degradation (ERAD) by the proteasome. It can, however, be argued that all disease-causing mutations lead to a conformational change to some extent, although not all are recognized as faulty by ERQC, e.g., gating mutations (*see Section 2.2. CFTR Mutations*) [17]. Checkpoints include a Hsp40/70/90 chaperone trap, which recognizes and removes the majority of misfolded CFTR protein [18], and a calnexin/calreticulin cycle involving the core N-glycosylated CFTR [19]. Before the newly produced protein can exit the ER and progress to the Golgi, folding is again assessed via arginine-framed tripeptide motifs (which should be hidden on the inside of the 3D protein structure) and di-acidic exit codes (which should be present on the outside) [20,21]. In the Golgi, the high-mannose glycosylation from the ER is modified to a more complex arrangement of sugars, containing galactose and fucose, among others [22]. At this point, CFTR is considered “mature” and delivered from the Golgi to the PM. Once at the PM, it is subject to peripheral quality control and repeated cycles of internalization and recycling [23,24]. For an extensive review on CFTR folding and trafficking, we refer to [16]. 

CFTR’s ion channel function starts with the unique R domain. This unstructured domain contains several important serine residues which, upon phosphorylation by protein kinase A (PKA), induce a conformational change [25]. This allows dimerization of the two NBDs, a first step towards the opening of the channel. Combined, the two NBDs create two ATP binding sites with the Walker A and B motifs from one domain connecting with the Walker C motif from the other [26]. The ATP-bound NBD dimer subsequently opens the channel (reviewed in [27]). The twelve transmembrane spans (TM) from the MSDs combine to form a pore allowing anions like chloride, bicarbonate but also iodide, to cross the PM. Only ATP binding site 2 has retained its ability to hydrolyze ATP, while site 1 was proposed to modulate the stability of CFTR function [28]. Dissociation of ADP in the NBD dimer after ATP hydrolysis at site 2 closes the channel (reviewed in [29]).

Over the last couple of years, cryo-electron microscopy (cryo-EM) has provided high-resolution structures of various states of both human and zebrafish CFTR, such as closed-open or (de)phosphorylated conformations, and with ATP or CFTR modulating compounds [30,31,32,33,34]. These structures have provided insights into the molecular mechanisms of CFTR channel function, i.e., its anion conductance pathway [35], as well as identified new features not previously described in ABC transporters, such as the N-terminal lasso motif [32]. This motif is of importance for CFTR channel gating as well as for folding of downstream domains and thereby might (partially) explain the molecular mechanism of several missense mutations located within this motif or in its vicinity [36,37,38]. Cryo-EM was further also used to determine the binding site of CFTR corrector VX-809 (*see Section 3.2.1. CFTR Correctors*) [34]. 

### 2.2. CFTR Mutations

When the *CFTR* gene and the major disease-causing mutation c.1521-1523delCTT (F508del; ~70% of CF alleles and found in ~85% of PwCF) were first described in 1989 [39,40,41], it was proposed that multiple mutations would make up the remaining 30% of CF alleles. To date, over 2100 *CFTR* mutations have been described (Cystic Fibrosis Mutation Database, https://www.genet.sickkids.on.ca/ (accessed on 10 May 2022)), of which currently 401 are confirmed to cause CF (Clinical and Functional Translation of CFTR (CFTR2), https://www.CFTR2.org/ (accessed on 10 May 2022)). Most mutations, however, are very rare, with only five mutations reaching an allele frequency above 1% and 20 above 0.3%. Compared to F508del, all other mutations are rare. An overview of the mutations discussed in this review can be found in Table 1.

Mutations in *CFTR* result in the loss or reduction of CFTR function and are commonly grouped by the mechanism by which this loss is caused, from Class I (most severe) to Class VI (mild) (Figure 1) [42,43]. PwCF with two mutations from the first three classes usually present with severe CF and are predominantly pancreatic insufficient. Mutations from Classes IV-VI give rise to a certain level of residual function, and are associated with milder forms of CF.

Class I contains the most severe mutations where no CFTR protein is produced. On the one hand, these cover mutations were the “blueprint” for CFTR is gone due to large insertions and deletions or by frameshift inducing indels. These mutations are unrescuable by means of small molecules (Class Ia, Figure 1), such as the 21 kb deletion CFTRdele2,3 (c.54-5940_273+10250del 21 kb; 0.3% of CF alleles) [44]. Canonical splicing mutations, for example 621+1G->T (c.489+1G>T) and 1717-1G->A (c.1585-1G>A) (both 0.9% of CF alleles), produce no normal *CFTR* mRNA [45], and are unlikely to be rescued by other means than gene therapy. 

This is in contrast to nonsense mutations (Class 1b) such as G542X (c.1624G>T; 2nd most common *CFTR* mutation, 2.5% of CF alleles) and W1282X (c.3846G>A; 5th most common mutation, 1.2% of CF alleles). Here, the *CFTR* mRNA is mostly degraded by the process of nonsense mediated mRNA decay (NMD) [46], and CFTR protein production is abolished. They are, however, considered rescuable by inhibition of NMD (*see Section 3.3.3. NMD Suppression*) and translational readthrough inducing drugs (TRIDs, *see Section 3.3.4. TRIDs*).

For Class II mutations, which include the most common F508del mutation as well as the 4th most common mutation N1303K (c.3909C>G; 1.6% of CF alleles), translation takes place but the immature protein does not achieve its native conformation. As a result, most misfolded proteins are recognized by the ERQC as faulty and primed for ER associated degradation (ERAD) by the proteasome. This results in a (near) absence of mutant CFTR channels at the PM [47]. Due to the location of the N1303K mutation near the end of the protein (in the NBD2 domain), folding is arrested at a late stage; indeed, while the N1303K folding intermediate resists ERAD, most of it accumulates instead in autolysosomes, where it is degraded via ER-associated autophagy [48].

In Class III, the mutant protein reaches the PM but its function is severely impaired. Generally speaking, the ATP-dependent gating of the channel is defective and, as a result, the open probability (P_O_) is strongly reduced, as is the case for G551D (c.1652G>A; 3rd most common mutation, 2.1% of CF alleles) [49].

Class IV mutations are situated around the channel pore, lowering its conductance of anions and thereby reducing the net transport of anions over the PM. R334W (c.1000C>T; 0.3% of CF alleles) belongs to this class. 

A reduced amount of correctly spliced *CFTR* mRNA is a hallmark of Class V mutations. These mutations cause cryptic splice sites, resulting in alternative splicing in a subset of transcripts, generating both aberrant and correct mRNA copies. One such example is 3849+10kbC>T (c.3717+12191C>T; 9th most common mutation, 0.8% of CF alleles). The remaining correctly spliced mRNA copies will give rise to wild-type (WT) CFTR proteins, but their overall quantity is significantly reduced [50]. 

A reduction in the number of CFTR channels is also found in Class VI [51], albeit by a different mechanism. Here, the protein stability is reduced compared to WT, enhancing its turnover and leaving fewer channels at the PM. Several C-terminal truncations, such as Q1412X (c.4234C>T), destabilize CFTR without interfering with its biogenesis [52]. Temperature rescued F508del, where cells are cultured at 26 °C to promote ERQC escape, is another example of this class [53].

Insights into the different molecular defect classes provide a starting point for the development of different therapeutic strategies needed to specifically tackle each defect. Not only does the variety and large number of *CFTR* mutations complicate causal treatment of CF, a single mutation can have multiple molecular defects—all of which need to be overcome to reverse the phenotype [54]. This phenomenon was best studied for F508del, which is usually quickly degraded (Class II) [55]. The small amount that reaches the PM shows defective gating (Class III) as well as reduced stability (Class VI) [56,57]. Multiple defects have since also been described for other mutations, including N1303K and W1282X (the latter, when NMD is suppressed) [58,59].

## 3. CFTR Causal Therapies

The need for distinct CFTR-rescuing strategies is underscored by the different *CFTR* mutation classes. In the next paragraph, we will discuss strategies that aim—alone or in combinations—to restore CFTR channel activity (Figure 2). They can be divided into three groups, based on whether they work on the protein, mRNA or DNA level. CFTR modulators are small molecules that interact directly with the CFTR protein. They either increase CFTR channel function (CFTR potentiators) or promote the folding and trafficking of CFTR to the PM (CFTR correctors). Other therapies target proteins that interact with CFTR to increase the amount of CFTR at the PM (proteostasis modulators and stabilizers) or its function (CFTR activators). mRNA-focused strategies aim to increase the amount of protein that is produced, either mutant or WT, and this treatment can further be supplemented with CFTR modulators where needed. As neither protein nor mRNA modulation provide long lasting effects, both need to be taken lifelong at regular intervals. Only when a correct genetic *CFTR* sequence is introduced into the host genome (by addition of a WT cDNA copy or correction of the mutated *CFTR* gene), the treatment has the potential to provide a cure for CF. 

We will start our discussion with protein targeting therapies and move towards the ultimate goal, i.e., to restore the CFTR “blueprint” in the DNA. Not co-incidentally, the CFTR-targeted drug discovery pipeline is following a similar route (Figure 2). Currently, CFTR modulator therapy is the only causal treatment for CF that is market-approved. It started in 2012, with the approval of Kalydeco™ for a small percentage of PwCF with the gating mutation G551D [60,61]. Its label was later extended [62,63], and currently Kalydeco™ is FDA approved for 97 *CFTR* mutations [64,65], covering ~8% of PwCF. In 2015 and 2018, CFTR modulator treatments Orkambi™ and Symdeko™ (Symkevi™ in Europe) were approved as the first causal therapies for PwCF homozygous for the F508del mutation [66,67,68,69]. Highly effective modulator therapy for PwCF carrying at least one F508del mutation only became available in 2019, marketed as Trikafta™ (Kaftrio™ in Europe, approved in 2020) [70,71].

While current CFTR modulator therapies have the potential to treat the majority of PwCF, about 15% of PwCF have mutations not responsive to (current) CFTR modulators [1]. These mainly include premature termination codon, frameshift and deletion mutations as well as certain canonical splice mutations (all in Class I—no protein) but also several missense mutations from Class II or III/IV that are refractory to the available CFTR modulators. As we will discuss below, several novel treatments are in development for the last 15% of PwCF currently without causal therapy. 

### 3.1. Improving CFTR Function: Activators, Potentiators and Co-Potentiators

#### 3.1.1. CFTR Activators

One might argue that restoring CFTR activity is most straightforward when CFTR is already present at the PM (Figure 3). This can be done by any process that enhances the availability of intracellular cAMP to phosphorylate and hence activate the CFTR channel, or by promoting its open state. Indeed, it was recognized early-on that enhancing the intracellular cAMP concentration by means of forskolin (adenylate cyclase activator) or IBMX (phosphodiesterase inhibitor) could enhance CFTR Cl^−^ currents [72]. Lubiprostone, approved for the treatment of chronic constipation [73], stimulates fluid secretion via activation of the CFTR channel through prostaglandin receptor EP4 mediated modulation of cAMP levels [74]. Recently, Shaughnessy and colleagues reported enhanced F508del rescue in CF airway epithelia when Trikafta™/Kaftrio™ was combined with lubiprostone [75]. The dual phosphodiesterase 3 and 4 inhibitor RPL554 (ensifentrine), in clinical development for chronic obstructive pulmonary disease (COPD) and asthma [76], was shown to activate CFTR with rare class III and IV mutations [77,78] and was tested in a small phase II trial with promising results ([79] & NCT02919995). Similarly, ATP analogues can activate the CFTR channel by locking it in an open state [80]. Several CFTR activators, including CBIQ (4-Chlorobenzo[F]isoquinoline), increased Cl^−^ secretion in heterologous cell models like Calu3 cells by simultaneous activation of CFTR and basolateral K^+^ channels [81]. 

A caveat for the use of CFTR activators is the generally unspecific nature of their mechanisms of action (MoA), which include interaction with general cellular functions such as kinase/phosphatase activity and ATP levels. Nevertheless, they have been highly valuable for unraveling CFTR regulation and gating mechanisms [82,83]. Forskolin and IMBX in particular are commonly used to activate CFTR channels in experimental settings, either to evaluate the function of CFTR mutants or to test therapeutic strategies. 

#### 3.1.2. CFTR Potentiators

In contrast to CFTR activators, CFTR potentiators interact directly with CFTR to increase the P_O_ of the channel while leaving the native regulation intact (Figure 3). The isoflavone genistein was one of the first CFTR potentiators discovered, already in the mid-nineties [84,85]. Although generally described as a tyrosine kinase inhibitor, it was shown to bind CFTR at two distinct sites, with opposite effects, creating a bell-shaped dose response curve [86,87]. A high affinity stimulatory binding site delays closing of the channel, thereby improving the P_O_ of CFTR [86]. A second, low affinity inhibitory binding site was proposed around the CFTR ATP binding sites, which could prevent binding of ATP and thus inhibit channel opening [87]. The exact binding sites of genistein have not yet been determined, although the high resolution of cryo-EM would potentially allow doing so [29]. A clinical trial (NCT00016744) testing genistein in combination with the proteostasis modulator sodium 4-phenylbutyrate (4PBA; *see Section 3.2.2. Proteostasis Modulators*) in people homozygous for the F508del mutation was eventually withdrawn as different classes of CFTR modulators with higher potential efficacy were being developed at the same time. 

Several of these novel CFTR potentiators were identified by high throughput screening (HTS). Phenylglycine (PG-01) and sulfonamide (SF-01) were identified as distinct chemical classes with CFTR potentiator activity by screening of 50,000 molecules [88]. As they were screened for their ability to potentiate F508del, mutant CFTR was first rescued by low temperature (27 °C) in order to allow measuring subsequent improved gating by halide sensitive yellow fluorescent protein quenching, indicative of improved CFTR function. As with genistein, there was synergy with cAMP agonists such as forskolin. While only PG-01 could potentiate CFTR mutants other than F508del, it was rapidly metabolized in vivo and therefore not clinically investigated [88]. Van Goor and colleagues screened 228,000 compounds using a fluorescence membrane potential assay on temperature-rescued F508del-overexpressing cells which, after chemical optimization, led to the discovery of VX-770 (ivacaftor) [89,90]. This compound was selected as a good clinical candidate based on its ability to potentiate multiple CFTR mutants, its high selectivity and promising pre-clinical pharmacokinetic profile. When evaluated in a clinical trial setting in PwCF carrying the gating mutation G551D, ivacaftor treatment resulted in rapid and substantial improvements in lung function (NCT00909532 and [61]). In 2012, ivacaftor monotherapy became the first approved CFTR modulator, marketed as Kalydeco™ [60]. Around the same time, VX-770’s mode of action was further unraveled, showing that it potentiates both mutant and WT-CFTR in a phosphorylation dependent but ATP independent manner [91], uncoupling the gating from the ATP hydrolysis cycle [92]. The exact binding site from which VX-770 exerts its effect remained elusive, however, and different strategies were employed to identify this site. NBD2 was dismissed, as VX-770 potentiated CFTR lacking this domain [93]. Byrnes and colleagues suggested the binding site at the intracellular loop 4 (ICL4), as this site was protected from hydrogen-deuterium exchange in the presence of VX-770 [94]. Cryo-EM on the other hand identified a binding site in the PM in a cleft created by TM 4, 5 and 8 [95]. Recently, both the ICL4 and TM binding site were confirmed using labeled VX-770 probes, suggesting there are in fact two binding sites [96]. Whether both sites contribute to CFTR potentiation, or one of these sites might contribute to the destabilization of CFTR in the presence of CFTR correctors [97], remains to be investigated further. 

A more stable variant of VX-770, VX-561 (formerly: CPT-561; also known as deutivacaftor), where 9 hydrogens are substituted by deuterium, is currently being tested in two phase 3 clinical trials (NCT05033080 & NCT05076149). Other potentiators, like H-01 and A-04 [98], P1, P2, P7 (from the Cystic Fibrosis Foundation’s CFTR Compound Program) and most notably ABBV-974 ([99]; formerly GLPG-1837) clustered together in combinatorial profiling, suggesting a similar mode-of-action as VX-770 [100]. In particular, VX-770 and ABBV-974 were shown to compete for the same binding site [101,102]. The efficacy of ABBV-974 on G551D potentiation was three-fold higher compared to VX-770, albeit with a lower potency [102]. In contrast to VX-770, however, ABBV-974 and other potentiators did not reduce the stability of the CFTR protein at the PM [98,103,104]. ABBV-974 was tested in clinical trials (NCT02690519 & NCT02707562; [105]), but has since been replaced by the novel potentiator ABBV-3067 (navocaftor) in the latest combination trial (with corrector ABBV-2222/GLPG-2222; NCT03969888). PTI-808 (also known as dirocaftor, NCT03500263, [106]) and QBW251 (icenticaftor; NCT02190604) were two other CFTR potentiators evaluated in clinical trials. While further development for CF is not planned, the latter is currently under investigation for the treatment of COPD, where mucociliary clearance is reduced due to smoke-induced acquired CFTR loss-of-function [107]. 

#### 3.1.3. CFTR Co-Potentiators

VX-770, alone or in combination with CFTR correctors, has provided substantial clinical benefit for PwCF. However, long-term follow-up in PwCF on Kalydeco™ has shown that lung function still declines while on the treatment, though at a slower pace [108]. Although many factors might contribute to the continued worsening of lung function, it has been shown for G551D and other (gating) mutations that VX-770 on its own is not able to fully restore the electrophysiological properties of mutant CFTR, such as P_O_, to WT levels [92,109]. This suggested that combinations of CFTR potentiators might further improve CFTR function. Indeed, combining VX-770 with other potentiators, such as genistein, synergistically improved CFTR functional rescue in gating mutants [100,110]. Moreover, it was recognized that potentiator combinations could be beneficial for mutations with a severe gating defect that are not rescued by current modulators, such as N1303K and W1282X [111,112,113]. 

The flavone apigenin, previously identified as a weak CFTR potentiator [84], was found in a synergy screen with ivacaftor and forskolin to effectively improve function of the W1282X truncation product, but only in the presence of VX-770 [113]. As it requires simultaneous VX-770 administration, apigenin was termed a co-potentiator or type II potentiator (in contrast to type I potentiators like VX-770 or ABBV-974) (Figure 3). Its MoA of co-potentiation has not been elucidated to date. Similarly, ASP-11 is a co-potentiator from the same class, equally showing great synergy with VX-770 and forskolin, in this case, for example, tested on processing and gating mutant N1303K [111,114]. Co-potentiators were later shown to benefit other NBD2 mutations besides N1303K, such as I1234del (c.3700A>G; the mutation gives rise to missense mutant I1234V as well as an alternatively spliced mutant with a six amino-acid deletion, I1234del), and the truncation products of W1282X and Q1313X (c.3937C>T)—all of which are currently without approved CFTR modulator therapy [112]. At maximum concentrations of VX-770, however, no additive or synergistic effect of ASP-11 was seen for F508del, suggesting a mutation or domain (NBD2) dependent mechanism [111]. 

VX-445 (elexacaftor), one of the CFTR modulators in Trikafta™/Kaftrio™, was recently shown to exert both potentiator and corrector activity (*see Section 3.2.1. CFTR Correctors*), depending on the mutation studied [114,115,116]. To date, the potentiator MoA of VX-445 has not been elucidated yet, although it showed synergy with both type I (e.g., VX-770) and type II (e.g., apigenin) potentiators [114,116]. This suggests the existence of at least three different potentiator MoA which can be targeted simultaneously for maximized CFTR rescue. 

### 3.2. Improving the Amount of CFTR at the Plasma Membrane: Correctors, Proteostasis Modulators and Stabilizers

Class II *CFTR* mutations present with maturation and trafficking defects, leading to a reduction in the number of channels at the PM. Therefore, folding of the mutant CFTR channel should be restored to levels that allow ERQC escape and trafficking to the PM. Alternatively, the degradation of misfolded CFTR by the ER or peripheral QC should be prevented by other means. Early-on, it was shown that incubation of cells at lower temperatures, e.g., 26 °C, prevented degradation of some class II mutations, most notably F508del [117], but not others, like G85E (c.254G>A) or N1303K [118,119]. F508del degradation is initiated early in the ER by the so-called chaperone trap [18,19], and lowering the temperature results in an accumulation of immature F508del protein by reduced proteasomal degradation, allowing F508del to escape the ERQC and be “rescued” to the PM [120]. Temperature sensitive mutants could also be rescued after treatment with 10% glycerol, which appeared to have a stabilizing effect on an early folding intermediate by acting as a chemical chaperone [121]. However, while they are useful tools to study CFTR (mis)processing, neither can serve as a potential therapeutic strategy. CFTR correctors, interacting directly with CFTR, or proteostasis modulators and stabilizers, targeting CFTR interactors, on the other hand, could be translated into therapeutic strategies. As the MoA of most compounds remain incompletely understood and compounds could potentially exert multiple mechanisms, the distinction between CFTR correctors, stabilizers and proteostasis modulators is not always as clear as presented here (Figure 4). 

#### 3.2.1. CFTR Correctors

Pharmacological chaperones are small molecules that, like the cellular chaperones present in the ER, assist in the folding of proteins. CFTR correctors can be considered pharmacological chaperones that, by direct interaction with the protein, improve the folding of mutant CFTR to such extent that early degradation is at least partially averted (Figure 5) [122,123]. Combinations of correctors can enhance the level of correction by combining different MoA [123,124]. While many CFTR activators and potentiators were rapidly identified, the discovery of CFTR correctors was off to a slower start, as the process of folding and trafficking involves many more components than its gating (reviewed in [125]). In addition, HTS assays have long relied mainly on functional CFTR read-outs that are more readily scaled to allow screening of large libraries [89,126,127]. This strategy assumed that CFTR, once rescued to the PM, is functional. F508del—and a number of other processing mutations—however, are known to also suffer from gating defects [54]. To account for this, screening assays looking directly at the trafficking of mutant CFTR proteins have since also been developed [128,129,130,131,132]. Recently, a fluorescence resonance energy transfer (FRET) based HTS platform was developed to evaluate the folding of nascent, ribosome-bound NBD1 mutants to specifically identify compounds able to restore the conformation of NBD1 [133]. 

Screening of 150,000 chemically diverse small molecules by Pedemonte and colleagues identified the bisaminomethylbithiazole corr-4a as a corrector able to improve F508del, but not N1303K, trafficking [126]. In addition, CFTR function was restored to a level similar to that of low temperature rescue in both heterologous and primary epithelial cell models [126]. Corr-4a was found to bind directly to CFTR [122], but the exact binding site and MoA have not been elucidated to date. Although initial studies suggested that corr-4a mainly stabilizes NBD2 [123], it appears that MSD2 is essential for corr-4a mediated rescue [134]. More recent studies also suggest its ability to stabilize MSD2 [135]. Corr-4a remains the archetypical example of type II correctors (i.e., targeting NBD2-MSD2) [123]. Other correctors of this type, such as the aminoarylthiazole derivative, FCG, were recently proposed to bind to NBD2 [136,137]. The MoA of type II correctors is complementary to that of other corrector types and hence allows to be added in conjunction (Figure 5) [120,123,138,139].

Another large screen of 164,000 synthetic compounds by Van Goor and colleagues identified the quinazoline compound VRT-325 as a corrector of F508del [89]. VRT-325 promoted the interaction between NBD1 and both MSD1 and MSD2 [140], and additionally could rescue folding mutants of the P-glycoprotein (a close family member of CFTR) [141]. At high concentrations, however, it was shown to inhibit CFTR function in a similar manner as the CFTR inhibitor Inh172 [142]. Another hit from the same screen, VRT-768, was optimized by extensive medicinal chemistry and structure activity analysis to VX-809 (lumacaftor) [143]. VX-809 was the first CFTR corrector that—together with potentiator VX-770—obtained market approval (Orkambi^TM^), as it significantly, though modestly, improved lung function in F508del homozygous PwCF [66]. Insights into VX809’s MoA were obtained from mutagenesis studies, where the limited additivity of VX-809 and revertant mutant R1070W suggested that both correct F508del folding via a similar mechanism, i.e., by restoring the NBD1:ICL4 interface [120,123]. At around the same time, it was shown that VX-809 is not able to correct the thermodynamic instability of the NBD1 domain [144], but rather seemed to stabilize MSD1 [38,145]. On the one hand, several studies have indeed showed that VX-809 assisted in compacting several transmembrane spans in MSD1 [146,147,148], although other studies have suggested that VX-809 interacts with NBD1 [149], both NBD1 and cytosolic loop 1 [150], or by stabilizing the protein once it is present at the PM [151]. By click-chemistry, VX-809 was shown to directly bind to CFTR [152], although the exact binding site and MoA of VX-809 were only recently elucidated by cryo-EM [34]. This method revealed that VX-809 fills an internal cavity which is intrinsically thermodynamically instable, formed by transmembrane spans 1, 2, 3 and 6 in MSD1. This finding is in line with other recent studies suggesting that VX-809 exerts its effects early in the biogenesis of CFTR [148], and might explain how VX-809 is also able to improve folding and trafficking of the WT-CFTR [47]. 

While VX-809 used to be the prototypical example of type I correctors and is still used in many experiments, it has now mainly been replaced in the clinic with VX-661 (tezacaftor), a molecule related to VX-809 which causes fewer side effects in PwCF and has an improved overall safety profile [69,70,71,153]. It shares its binding site and MoA with VX-809 [34]. Tezacaftor is market-approved in combination with potentiator VX-770 (Symdeko™/Symkevi™) or VX-770 and corrector/potentiator VX-445 (Trikafta™/Kaftrio™). Other type I correctors, such as C18 (VRT-534; from the Cystic Fibrosis Foundation’s CFTR Compound Program), ABBV-2222 (formerly GLPG-2222; also known as galicaftor), FDL-169 and trimethylangelicin (derivatives) rescue CFTR through a similar MoA [123,138,154,155]. ABBV-2222 was tested in clinical trials and was shown to be well-tolerated, but no change in lung function was observed ([156] & NCT03045523, NCT03119649). New trials combining ABBV-2222 with the next generation potentiator ABBV-3067 are currently ongoing (NCT03969888). More recently, a novel type I corrector, ARN23765, was described with picomolar potency (EC_50_: 38 pM)—which showed prolonged CFTR rescuing effects in vitro (>36 h) at concentrations 5000 times lower compared to VX-809 [157]. 

As type I and II correctors or combinations were unable to rescue all aspects of the F508del folding defect, in particular the thermodynamic instability of NBD1, the search for novel corrector types continued. Screening of 600,000 compounds identified one of the first type III correctors, 4172, which was proposed to bind and stabilize NBD1 [138,139]. It shares a corrector MoA with the next generation correctors VX-445 (elexacaftor; part of the triple combination Trikafta™/Kaftrio™) [138,158] and VX-659 [159]. While the exact binding site and corrector MoA of VX-445 remain unknown, several studies suggest a binding site located at the NBD1 [138,160]. Importantly, VX-445 could at least partially prevent thermal unfolding of NBD1, the hallmark of type III correctors [138]. 

Several other correctors, whose MoA are not covered in corrector types I-II-III or have yet to be elucidated, have also been described. One of the first CFTR correctors to be identified, the benzo(c)quinolizinium compounds (MPBs) were thought to bind NBD1 and promote maturation as well as stimulate function [161,162,163]. Bis-phosphinic acid derivatives c407 and G1, targeting the NBD1, were recently modeled to fill the cavity left by the missing phenylalanine side chain in F508del, thereby restoring the NBD1:ICL4 interface [164,165]. In cell models, additivity of c407 with VX-809 suggests these compounds might represent a complementary MoA with existing CFTR corrector types. Other pharmacological chaperones targeting NBD1 are phenylhydrazones [166], such as RDR-1 [130], which enhanced CFTR currents in differentiated human bronchial epithelial cell (HBE) cultures synergistically with VX-809 and MCG1516A [167], and which, since then, have been proposed to bind at the NBD1:NBD2 interface [168].

#### 3.2.2. Proteostasis Modulators

Proteostasis modulation has long been considered for the causal treatment of CF as a means of rescuing mutant trafficking to the PM (Figure 4). It is an umbrella term for a diverse set of strategies that aim to indirectly rescue mutant CFTR to the PM, by interacting with proteins involved in (mutant) CFTR degradation. One of the first compounds that was found to improve the expression of F508del at the PM was 4PBA [169], shown to increase the expression of pro-folding chaperones like Hsp90 and ERp29 [170,171]. Overexpression of ERp29 rescued F508del to the PM in bronchial epithelial IB3-1 cells, while knockdown decreased levels of PM WT-CFTR [170]. A pilot clinical trial showed partial restoration of nasal epithelial CFTR function, but no reduction in the concentration of sweat chloride [172]. It was later tested in combination with genistein as well (NCT00016744, *see Section 3.1.2. CFTR Potentiators*). Currently, an optimized pro-drug variant of 4PBA, glycerol phenylbutyrate (GPBA), is being tested in a phase I/II study (NCT02323100). Additional ER chaperones were identified whose modulation could rescue F508del trafficking. Proteomics and differential interactomics between WT-CFTR and F508del revealed inhibition of Hsp90 co-chaperones Aha1 [173] or PTPLAD1 [174] as potential novel therapeutic strategies. Another way of altering the expression of ER chaperones is by inhibiting specific histone deacetylases (HDACs). This was studied, for example, by inhibiting HDAC7 using HDAC inhibitors (HDACi), such as suberoylanilide hydroxamic acid (SAHA, vorinostat) or by siRNA knockdown, both of which improved the PM expression and activity of F508del [175]. Other HDACi, such as the fungal metabolite apidicin, showed improved F508del rescuing potential, although they were less potent at HDAC inhibition [176]. Several HDACi have also been shown to rescue a panel of rare CFTR mutants, in particular class II mutations P67L (c.200C>T) and E92K (c.274G>A) [177]. Their biggest caveat, it seems, is that CFTR rescue by HDAC modulation appears highly cell-type-dependent. For example, their rescuing effects in the immortalized CFBE41o- lung cell line, where F508del overexpression is driven by a viral promoter, could not be confirmed in primary airway cells [178,179]. Other primary cell types remain to be investigated though. 

Besides modulating the expression of CFTR chaperones, another strategy consists of disrupting the calnexin/calreticulin ERQC checkpoint. The α-1,2-glucosidase inhibitor miglustat (n-butyldeoxynojirimycin), which is used to treat type I Gaucher and Niemann-Pick type C disease [180], increased F508del trafficking via the inhibition of the deglycosylation of nascent proteins, which usually takes place at this checkpoint [181]. Although F508del function was rescued in airway epithelial cells and CF mice, no clinical benefits were observed in a phase II trial of miglustat in F508del homozygous PwCF [182]. The small molecule thapsigargin has been shown to disrupt the interaction between F508del and calnexin by depleting ER Ca^2+^, which rescued the nasal epithelial potential defect in F508del mice [183]. Roscovitine (seliciclib) increased F508del trafficking by a similar MoA, as well as by inhibiting proteasomal activity, thus preventing ERAD [184]. Moreover, it was shown to have additional favorable properties for PwCF, as it displays anti-inflammatory properties, such as enhanced neutrophil clearance and reduced eosinophil degranulation, through its inhibition of cyclin-dependent kinases (CDKs) ([185]; reviewed in [186]). When tested in a Phase II trial (ROSCO-CF, NCT02649751), unfortunately, no significant improvements were found in spirometry, inflammation, infection or sweat chloride [187]. In the meantime, structural roscovitine-analogues have been synthetized, which were recently found to also restore trafficking of other proteins similar to CFTR, such as ABCB4, in cell models [188]. 

Curcumin is a natural product which was first described to correct F508del trafficking via its interaction with the ER-located calcium pump (SERCA) [189]. However, later studies could not confirm this rescue [190,191]. As it is structurally related to CFTR potentiator genistein, it was consequently hypothesized that, like genistein, it could also bind directly to CFTR and increase its P_O_ [192]. Recently, a small clinical study investigated the combination of curcumin and genistein in PwCF carrying at least one S1251N (c.3752G>A) gating mutation (NTR4585 & [193]). While some functional rescue was measured in rectal organoids of the respective involved PwCF, it was low compared to VX-770 treatment [193]. Clinically, no clear effect was observed, although it was argued that the plasma concentrations measured for both compounds were much lower than those typically used in in vitro studies. 

Another investigated pathway to rescue trafficking of misfolded CFTR, is to interfere with the unfolded protein response (UPR). As such, latonduine, a sponge alkaloid, and related analogues identified by phenotypic screening, were reported to inhibit the activity of poly-ADP ribose polymerases (PARP) 3 and 16 [194]. These regulate UPR via ADP-ribosylation of IRE-1-α and PERK, two ER stress sensors that initiate UPR. Inhibition of UPR by PARP3 and PARP16 rescued F508del trafficking in cell models [194,195,196]. Misfolded CFTR proteins are “flagged” for degradation by ubiquitination. Preventing ubiquitination thus presents another means for enhancing ER escape and trafficking. In that light, the E1 ubiquitin-activating enzyme could be inhibited by PYR-41, a small molecule which synergistically with corrector C18 rescued F508del trafficking and function, as by itself it only increased the amount of immature, ER-localized protein [197]. This compound was further chemically optimized into “7134” in order to reduce toxicity while maintaining its ability to augment F508del function when co-treated with VX-809 [198]. Recently, Borgo and colleagues used the small molecule inhibitor TAK-243 (also known as mln7243) of the ubiquitin-activating enzyme UBA1 to enhance the efficacy of Trikafta™/Kaftrio™ treatment in primary airway epithelial cells [199]. This effect was not limited to F508del, but also translated towards other processing mutations, including the difficult to rescue mutation N1303K. Alternatively, preventing ubiquitination by inhibiting the ubiquitin ligase RNF5, rescued F508del in a primary airway epithelial cell model [200]. As another approach, de-ubiquitination targeting chimeras (DUBTAC) can be used to recruit de-ubiquitinases and reverse ubiquitination of misfolded proteins [201]. For CF it was shown that F508del could be stabilized in epithelial cells by linking the OTUB1 de-ubiquitinase recruiter EN523 to VX-809—which conferred specificity to CFTR—and allowed trafficking to the PM and subsequent potentiation by VX-770 [201]. Small Ubiquitin Like Modifier (SUMO) conjugation, SUMOylation, is a post-translational modification which can regulate a vast number of cell processes, including protein degradation [202]. Prevention of SUMOylation through SUMO2/3 by overexpression of activated STAT isoform 4 (PIAS4), slowed down the degradation of immature F508del-CFTR as well as several other processing mutants [203,204]. Improved folding (i.e., increased mature/immature CFTR ratio on Western blot analysis) was also obtained for a panel of rare mutations, including N1303K, when PIAS4 overexpression was combined with the type I corrector C18 [203]. For a more extensive discussion of ubiquitination as a target for CFTR rescue, we refer to [205], but in summary, any strategy interfering with this process is able to promote the escape of misfolded CFTR, though predominantly in conjunction with correctors does this lead to improved CFTR function. 

Moving further in the biogenesis cycle of CFTR, enhancing autophagy is another strategy proposed to enhance expression of mutant CFTR at the PM [206]. In that context, cysteamine, approved for the treatment of cystinosis [207], was tested in CF mice, followed by a small open-label clinical trial in PwCF, where it was shown to reduce sweat chloride levels in F508del carrying PwCF, but not in a cohort which contained PwCF carrying two non-F508del mutations not responsive to current CFTR modulators [208,209]. More recently, however, functional rescue by cysteamine and thymosin α1—the rescue by the latter also reported to occur through promoting autophagy [210]—could not be confirmed by several other labs [211,212,213], dampening the enthusiasm to stimulate autophagy as a way of reversing the CF phenotype. 

Phosphodiesterase type 5 (PDE5) inhibitors cause vasodilation via smooth muscle relaxation by inhibiting cGMP degradation [214]. PDE5 inhibitor sildenafil and structural analogues, such as vardenafil and KM11060, were reported to enhance CFTR maturation, i.e., the fully glycosylated “band C” on Western blot, PM expression and function [215,216,217]. Although the MoA by which maturation and PM expression are rescued by these compounds remains unclear, it is possible that its phosphodiesterase modulation also activates CFTR channels [216,218]. Since sildenafil was already FDA approved for erectile dysfunction and pulmonary arterial hypertension [214], it could relatively easily be repurposed to PwCF. Although the therapy was safe and well-tolerated, no clear clinical benefit was observed (NCT00659529, NCT01132482, [219]). Similarly, the soluble guanylate cyclase stimulator riociguat, which is also approved (as Adempas™) for the treatment of pulmonary hypertension [220], modulates the same pathway as the above phosphodiesterase inhibitors [221]. However, while CFTR processing and function was increased in vitro and in vivo in F508del mice, it did not improve CFTR activity or lung function in the Rio-CF study (NCT02170025), a small clinical trial in F508del PwCF [221,222]. 

A final example of proteostasis modulators is the non-steroidal anti-inflammatory drugs (NSAIDs) such as ibuprofen and glafenine. These block the production of prostaglandins and thromboxanes by inhibiting the cyclooxygenase (COX) enzymes that convert arachidonic acid to prostaglandins [223]. They were also reported to rescue CFTR expression [132,224]. Recently, Carlile and colleagues showed that glafenine and analogues corrected misfolded class II CFTR by preventing the stimulation of the prostaglandin E2 receptor PE4, which was confirmed by siRNA mediated knockdown of PE4 [225]. It was further proposed that targeting this metabolic pathway, rather than specific protein production/QC pathways, could potentially rescue misfolded proteins other than CFTR as well [225].

In summary, the search for proteostasis modulators able to promote mutant CFTR trafficking to the PM has been extensive. As evidenced by the numerous examples discussed in this section, many different strategies have been investigated. However, none of them have translated so far into a therapy for PwCF. The main reasons for this are the occurrence of cell-type specific rescue, often not translating to rescue in primary cells, and the lack of clinical benefit in PwCF in clinical trials. This highlights how the choice of cell model greatly influences the degree of CFTR rescue, in particular when rescued via non-direct mechanisms such as proteostasis modulation. Any potential strategy (including, but not limited to, proteostasis modulation) should thus be tested in primary cell models with good translational value as soon as possible, to prevent attrition at later stages.

#### 3.2.3. Stabilizers

As CFTR has now reached the PM, with or without the help of CFTR correctors and proteostasis modulators, the next important step is to make sure it stays there. Once at the PM, CFTR is continuously endocytosed and recycled back to the PM [16]. During this process, misfolded proteins, however, are recognized by the peripheral quality control (PQC) and, rather than recycled, are instead ubiquitinated to prime them for lysosomal degradation, reducing the number of available CFTR channels [16,23]. Stabilizers thus aim to improve the amount of CFTR channels at the PM by stabilizing the CFTR protein and slowing down its turn-over, a strategy which is complementary to CFTR correctors or proteostasis modulation (Figure 4). As discussed earlier, it is not always possible to discriminate between CFTR correctors, proteostasis modulators and stabilizers, especially as not all the mechanisms of these compounds have been elucidated in detail yet. Compared to correctors and proteostasis modulators, only a few compounds have been developed to specifically target peripheral stability of the CFTR protein. The only CFTR stabilizer tested in clinical trials to date is cavosonstat (N91115), an inhibitor of the S-nitrosoglutathione reductase (SNGOR) ([226], NCT02275936, NCT02013388, NCT02500667). This compound exerts its CFTR stabilizing function by decreasing the internalization rates of CFTR and thus extending its half-life at the PM [227,228]. Specifically, this is achieved through interfering with the PQC. Cavosonstat preserves S-nitrosothiols by inhibiting SNGOR, which results in the nitrosylation of Hsp70/Hsp90 organizing protein (HOP). This, in turn, lowers the interaction of CFTR with CHIP (C-terminus of Hsp70 interacting protein), thereby preventing its degradation. Although the Phase I trial was not powered to evaluate efficacy, no effects were observed with mono-therapy of cavosonstat in F508del homozygous PwCF, besides a small reduction in sweat chloride [226]. In a Phase II follow-up study where cavosonstat was administered together with ivacaftor and lumacaftor, no clinical benefits were observed [229]. 

In addition, CFTR modulators have been found to influence CFTR stability. As such, the corrector VX-809 and other type I correctors were shown to not only correct folding in the ER, but also improve stability once at the PM [151]. In contrast, potentiator VX-770 destabilizes corrector type I-rescued CFTR [97], even in the presence of VX-445 [230]. Other potentiators with similar mechanisms as VX-770 do not destabilize the protein, so the choice of combination of small molecules can affect protein stability [98,230]. Alternatively, the destabilization could be prevented by co-treatment with the hepatocyte growth factor (HGF) [231,232], which stabilizes CFTR by promoting its anchoring to the actin cytoskeleton via the Rac1 GTPase [232,233]. The vaso-active intestinal peptide (VIP) has been shown to increase the membrane localization by promoting interaction between CFTR and the N^+^/H^+^ exchanger regulatory factor 1 (NHERF1) and ezrin/radixin/moesin (ERM) complex, while at the same time inhibiting interaction with the CFTR associated ligand (CAL) [234]. Activation of the cAMP sensor EPAC1 also promoted the interaction between CFTR and NHERF1, stabilizing CFTR at the PM [235,236]. Inhibition of the protease calpain 1, on the other hand, stabilized CFTR through promoting its interaction with ezrin [237]. Other strategies have also focused on these interaction partners, by overexpressing NHERF1 [238], inhibiting CAL [239], or by preventing endocytosis [240,241] and promoting recycling (reviewed in [242]). CFTR PM levels have furthermore been shown to be regulated through phosphorylation of Y512 by the spleen tyrosine kinase SYK [243]. Consequently, inhibition of SYK or the adaptor SHC1, which recognizes phosphorylated Y512-CFTR, increased the amount of PM CFTR [244]. 

In conclusion, stabilizing CFTR at the PM is, in theory, an attractive way to increase CFTR PM density. However, all specific stabilizing strategies are still in early pre-clinical development. Therefore, on the short term, stabilization of CFTR is most likely to occur from optimizing CFTR modulator combinations to avoid destabilization.

### 3.3. Improving the Amount of Immature CFTR Protein

So far, all strategies discussed have focused on modulating the mutated CFTR protein to, at least, a certain level of CFTR function. However, these strategies assume there is sufficient immature protein available to correct. In the next section, we will focus on strategies that aim to enhance the amount of immature protein by interacting with the *CFTR* mRNA and the process of translation rather than to modulate the mutant protein itself. The new pool of available immature protein can then be rescued further by combining it with CFTR correctors and potentiators as needed. In certain cases, however, it might suffice to modulate the translation process in the ER to overcome CFTR defects. By reducing ribosome velocity during translation for example through suppression of the ribosomal protein L12 (RPL12), mutant CFTR trafficking and function has been shown to be partially restored [245,246]. 

In the next paragraphs, we will discuss other means of increasing the amount of immature protein. First, we will detail the development of amplifiers and miRNA modulation, which could be of use to rescue *CFTR* mutations where protein is still produced, i.e., class II-VI. Second, we will focus on NMD suppression and TRIDs, potential therapies for PwCF with pre-termination codon (PTC) mutations.

#### 3.3.1. Amplifiers

An HTS of 54,000 compounds performed by Giuliano and colleagues identified a novel CFTR modulator, which was neither a potentiator nor a corrector but showed synergism with both [247,248]. This modulator, PTI-CH, termed an amplifier, increased the amount of immature CFTR by stabilizing the mRNA, which resulted in a 1.5-2-fold increase in available *CFTR* mRNA transcripts [248]. This strategy could be of particular interest for Class II mutations, as it would increase the pool of immature protein available for corrector rescue, as well as for Class V mutations, where, due to cryptic splice sites only a fraction of the *CFTR* mRNA is correctly spliced (Figure 6). PTI-CH was, furthermore, found to act co-translationally by promoting translation and insertion of the first transmembrane spans into the ER, a process which is inherently inefficient [249]. PTI-CH was shown to bind directly to the poly(RC) binding protein 1 (PCBP1), whose interaction increased *CFTR* mRNA in the ER. This interaction appeared specific for the *CFTR* mRNA [249], though in a mutation-agnostic manner, as it was shown to stabilize both F508del, I1234V and WT mRNA [248,250]. Based on these promising results, the optimized amplifier PTI-428 (nesolicaftor), was tested together with potentiator PTI-808 and corrector PTI-801, in a phase 1/2 clinical trial in PwCF carrying F508del alleles (NCT03500263). On average, an improvement in lung function of 8% was observed [251]. While the absolute improvement was lower compared to that of Trikafta™/Kaftrio™, rescue was highest in subgroups with the highest disease burden (+10–12%) or with poor response to previous CFTR modulators (+12%). Recently, these compounds (including the amplifier) were licensed so that development may continue [252].

#### 3.3.2. miRNA Modulation

Post-transcriptional regulation of *CFTR* mRNA by microRNAs (miRNAs) may affect the number of available transcripts, as exemplified in the CF airway epithelium (reviewed in [253,254]). Among the miRNAs upregulated in PwCF, several were found to negatively regulate *CFTR* expression by interacting with the 3′ untranslated region (UTR) of the *CFTR* mRNA, including miR-145-5p and miR-509-3p [255,256]. Inhibition of these miRNAs with peptide nucleic acids (PNA) directed against them resulted in increased expression of WT or F508del CFTR [257,258,259]. However, these strategies do not only inhibit the interaction between the miRNA and *CFTR* template, but also its interaction with other targets, potentially causing unwanted side-effects. To overcome this issue, De Santi and colleagues focused on the development of a *CFTR-*specific approach by blocking specific miRNA binding sites in the *CFTR* 3′UTR, the so-called target site blockers (TSBs) [260]. They showed that treatment with TSBs significantly enhanced rescue of F508del by VX-770 and VX-809/VX-661 in a CF bronchial epithelial cell line. 

#### 3.3.3. Nonsense Mediated mRNA Decay (NMD) Suppression

The majority of PwCF who are currently not eligible for causal CF therapies have one or two nonsense *CFTR* mutations. Approximately 5% of all CF alleles in the CFTR2 database are nonsense mutations, which create PTCs, giving rise to truncated proteins but also, in many cases, which prime the *CFTR* mRNA for NMD (Figure 7; left panel). For a review on NMD in health and disease, we refer to [261]. A PTC can trigger NMD when it generates long 3′UTRs or when exon junction complexes (EJC) are formed downstream of the PTC. NMD reduces the amount of mutant *CFTR* mRNA and hampers subsequent rescue of the CFTR protein by other means, such as by TRIDs (*discussed next*) and by CFTR modulators [262,263]. The level of NMD differs not only between cell-types, but also depends on the specific PTC mutation. In addition to this, individual variation is at play, as seen between PwCF with the same genotype [46,262,264,265]. Clarke and colleagues reported that in primary nasal epithelial cells from 10 PwCF carrying the G542X/F508del genotype, mRNA expression of the G542X alleles was reduced by ~60% [264]. Differences in the abundance of the 3′ and 5′ of the *CFTR* mRNA suggest that some of the mRNA is partially degraded and not a suitable substrate for full-length protein translation [46]. Making sure that a sufficient amount of suitable *CFTR* mRNA is available is therefore the first important step towards effective treatments for PTC-induced CF (Figure 7; right panel). Several strategies have been proposed in pre-clinical models to prevent NMD and enhance mRNA expression of PTC *CFTR* (reviewed in [266,267]). Inhibition of the NMD activator serine/threonine-protein kinase SMG1, via the small molecule inhibitor SMG1i, resulted in increased levels of G542X, W1282X and other PTC containing mRNAs in various models, including airway epithelial cells, primary rectal organoids and several CF animal models [58,268,269,270,271,272,273,274]. As SMG1 is active in other cellular processes besides NMD, its inhibition causes considerable toxicity, which likely precludes it from translation into a therapy [272,275]. Similarly, siRNA mediated knockdown of the regulator of nonsense transcripts, UPF1—which gets phosphorylated by SMG1 [276]—increased mRNA levels of exon 22 PTC mutations, increasing their overall response to other therapies [277]. In addition, inhibition of the interaction between UPF1 and SMG7 by small molecule NMDI-14, suppressed NMD and stabilized W1282X mRNA [263]. Some compounds, like the clinically approved drugs amlexanox and escin were suggested to provide dual NMD and PTC suppression [272,278,279]. All of these treatments, however, inhibit the NMD machinery in a non-*CFTR* specific manner, thereby posing potential unwanted side-effects related to unspecific NMD suppression. 

A more targeted approach can therefore be achieved by designing antisense oligonucleotides (ASO) that bind the *CFTR* mRNA and prevent EJC deposition downstream of the PTC [280]. Alternatively, removal of the PTC from the mRNA allows the prevention of NMD. Both strategies were shown to rescue the W1282X mutation either by removing the gene downstream of the W1282X mutation (and thus removing the remaining EJCs) by means of clustered regularly interspaced short palindromic repeats (CRISPR)/Cas9 gene editing [281] or by inducing skipping of exon 23 (CFTRex23del), in which W1282X is located, using ASOs [282,283,284]. Exon 23 is an in-frame exon [284], meaning that no additional mutations or frameshifts are introduced by the deletion of this exon. While CFTRex23del is hardly functional on its own, it can be rescued by CFTR modulators [282,283,284]. Whether this exon skipping strategy can also be applied to other PTC mutations outside of exon 23, remains to be investigated. In general, suppression of NMD will need to be paired with other strategies, such as TRIDs (*see next section*) and CFTR modulators, to reach sufficient levels of CFTR function (Figure 7, right panel).

#### 3.3.4. Translational Readthrough Inducing Drugs (TRIDs)

Saving the PTC mRNA molecules from degradation, however, is only the first hurdle to restoring the function of most PTC CFTR mutants. The PTC will still give rise to a truncated protein, which, in most cases, is not functional or responsive to CFTR modulators. Suppressing this premature translation termination would therefore be highly favorable. Aminoglycoside antibiotics such as gentamicin and G418 (geneticin) have been shown to promote translational read-through via near-cognate mispairing of an aminoacyl-tRNA with the PTC for several *CFTR* mutations in cell models and a G542X mouse model [285,286,287]. Trials in PwCF showed variable responses to treatment with aminoglycosides without prolonged clinical benefit [288,289,290]. In G542X mice, the aminoglycoside amikacin was more effective at suppressing premature translation termination compared to gentamicin [291]. However, rescue by tobramycin, another aminoglycoside antibiotic commonly prescribed in CF, was much more modest [287,292]. 

The oxadiazole PTC124 (ataluren) was developed as a non-aminoglycoside alternative with TRID properties that promotes ribosomal readthrough of PTCs, but not normal termination codons [293]. This molecule was discovered in a generic HTS using a PTC-containing luciferase reporter. Currently, it is EMA approved to treat nonsense forms of Duchenne muscular dystrophy [294], but it was quickly also tested for treatment of PTC mutations causing CF [295]. While early clinical studies suggested a beneficial effect of PTC124 ([296,297,298] and NCT00237380, NCT00351078, NCT00458341), ultimately no significant effect on lung function was observed in a Phase III trial ([299] and NCT00803205). Interestingly, in rectal organoids of human or mouse origin, PTC124 also failed to rescue CFTR function [300,301]. Other oxadiazoles were shown to be more efficient than PTC124 in a Fisher rat thyroid (FRT) cell model [302,303], but for now, none have yet reached clinical testing.

Novel, synthetic aminoglycosides, such as NB54 and NB124 (ELX-02), had a more favorable toxicity and bio-availability profile compared to first generation aminoglycoside TRIDs [304,305,306]. The most promising TRID currently in clinical evaluation is ELX-02 [307]. Phase I trials have shown its safety ([308] and NCT0280796, NCT03292302), and currently, Phase II trials are ongoing to evaluate its efficacy (NCT04126473 and NCT04135495). In contrast to PTC124, ELX-02 has been shown to restore CFTR function in G542X human rectal organoids [309]. Moreover, it was able to rescue several other PTC mutations in isogenic human bronchial cell lines (16HBEge) [268]. Here, ELX-02 was combined with several other treatments, including NMD suppressors, amplifiers, correctors and potentiators, to reach maximum functional rescue. 

Indeed, preventing premature translation termination does not necessarily fix the resulting protein. Instead, it will often alter the nonsense mutation into a missense mutation, in the case that a non-native amino-acid is incorporated [310]. Which amino-acid is incorporated depends not only on the identity of the PTC, but also on the sequence context in which it resides. For example, while G542X and W1282X are both UGA PTCs, different amino-acids were found to be incorporated after treatment with TRIDs [310]. Similarly, G542X appears to be more responsive to TRIDS than W1282X due to differences in the respective sequence contexts [311]. To rescue the resulting missense mutations, many pre-clinical studies now focus on combining TRIDs with other classes of compounds, which in most cases synergistically improves functional rescue [58,263,268,269,270,271,272,273,274,277].

Recently, a HTS of >750,000 compounds identified SRI-37240 and its more potent derivative SRI-4131 as novel TRIDs with a mechanism that is complementary to that of aminoglycosides and oxadiazoles [312]. These compounds prolong the translational pause at the PTC position in the transcript by reducing the abundance of translation termination factor eRF1, thereby promoting readthrough. Additionally, another HTS 1536-well platform to phenotypically screen specifically for G542X readthrough and subsequent rescue by CFTR modulators was recently published [313]. Other novel strategies have focused in particular on restoring native amino-acid incorporation. Using anticodon engineered (ACE)-tRNAs, this could indeed be achieved for G542X, R1162X (c.3484C>T) and W1282X without disrupting physiological termination codons [314,315].

### 3.4. Producing Correct CFTR

The previously discussed strategies have focused on enhancing CFTR production, trafficking, function and stabilization. Alternatively, rather than repairing an “impaired” CFTR protein, one might circumvent this by providing either a new “blueprint” to generate WT-CFTR or by repairing the mutation on the mRNA or DNA level (Figure 8). The final strategies we will discuss in this review will focus on restoring the production of WT-CFTR as a causal treatment for CF. 

### 3.4.1. mRNA Repair

Two decades ago, it was shown in a minigene model and in F508del bronchial epithelial cells that the *CFTR* pre-mRNA could be repaired by the use of spliceosome-mediated RNA *trans-*splicing (SMaRT) with a super-exon for exons 11–27 (10–24 in the old nomenclature) [316,317]. The super-exon contains multiple exons and replaces the sequence downstream of the *trans-*splice site on the mRNA, inserting the CUU that is missing in F508del. While this repair was aimed at F508del, it could be extrapolated to other mutations located in exons 11–27. A duplex of two short single complementary-strand oligonucleotides that were constructed to repair the F508del mutation also led to phenotypic recovery of channel currents in F508del overexpressing cells [318]. One of these oligos was further developed into eluforsen (QR-010), a 33 nt ASO, which restored F508del-CFTR function in a differentiated airway epithelium and in F508del mice [319]. The mechanism by which the antisense oligo corrected the F508del mRNA remains unknown. Eluforsen was administered via inhalation in two small trials to F508del homozygous PwCF (NCT02532764 & NCT02564354). While treatment was well tolerated and the respiratory symptom score (CFQ-R RSS) and CFTR currents, as measured by nasal potential difference (NPD), improved, no significant change in lung function was obtained four weeks after treatment [320,321], leading to a halt in further clinical development to date. 

mRNA repair strategies have not solely focused on F508del, though. Using an engineered site-directed RNA editase, the W496X (c.1487G>A) mRNA expressed in *Xenopus* oocytes, could be corrected [322]. More recently, Melfi and colleagues edited the W1282X mRNA in cell lines using “REPAIRv2” (RNA Editing for Programmable A to I Replacement, version 2), a CRISPR/dCas13b-based molecular tool which allows mRNA editing of adenosines to inosines, which are read as guanosines [323]. However, the editing efficiency would need to be further optimized in order to reach sufficient rescue. 

Class V *CFTR* mutations cause cryptic splice sites, which lead to the insertion of additional nucleotides into the mRNA. Although a fraction of the *CFTR* mRNA is correctly spliced and creates WT-CFTR channels, this is usually not enough to ameliorate CF symptoms. Restoring the correct splicing of these mutations would therefore be a promising therapeutic approach as well. Splice altering ASOs are already used in the clinic, for example to treat spinal muscle atrophy (nusinersen/Spinraza™) [324]. In CF, splice altering ASOs were shown to correct the 2789+5G>A (c.2657+5G>A) [325] and 3849+10kbC>T [326,327,328] mutations. The clinical candidate ASO SPL84-23 was able to rescue CFTR expression and function in primary bronchial and nasal epithelial cells to ~40% of WT [328], received orphan drug designation recently by both the FDA and EMA, and a first clinical trial is planned for the end of 2022 [329].

### 3.4.2. mRNA Therapy

One alternative to mRNA editing is to bypass the mutated mRNA altogether and provide a WT-transcript instead. In this way, by producing WT-CFTR, the CF phenotype can be reversed. As this approach is mutation agnostic, e.g., not dependent on the *CFTR* genotype, it can potentially be used for all PwCF. Delivery of the mRNA molecules to the right cells, however, is one of the major challenges for mRNA replacement therapy in CF and in general [330]. Besides the need to be safe, reliable and efficient, the mRNA should be stable enough to allow sufficient translation [331]. For an in-depth discussion on mRNA delivery we refer to [330,331]. The use of mRNA in medicine has taken a tremendous flight with the implementation of mRNA vaccines against SARS-CoV-2 (reviewed in [332]). For CF, the first-in-man therapeutic mRNA trial (RESTORE-CF) is currently underway (NCT03375047). MRT5005 is a WT, modified *CFTR* mRNA that is provided to airway cells via nebulization [333,334]. The most recent interim results suggest that multiple doses of the mRNA treatment are safe and well tolerated, although at this stage no change in lung function was observed [333]. The mRNA sequence and lipid nanoparticle delivery vehicle have received further optimization and are planned to be tested in a clinical trial in the near future as a next generation mRNA therapeutic [333]. ARCT-032 (LUNAR-CF), another nebulized mRNA replacement therapy, is also in preparation for clinical testing [335]. Recent preclinical work showed that chitosan nanoparticles, loaded with both the *WT*-*CFTR* mRNA and capsaicin, the latter blocking the epithelial sodium channel ENaC that is hyperactive in CF, could normalize both CFTR and ENaC function [336]. The novelty of this strategy lies in its ability to target two CF-related defects simultaneously. 

### 3.4.3. Targeting the DNA—Towards a Cure for CF

Up to now, we have discussed strategies targeting either the CFTR protein or its mRNA. However, none of these therapies will be able to cure CF. As long as the underlying defect, e.g., the mutation within the *CFTR* gene, remains present, so will the disease. If protein or mRNA therapies are suspended, the disease will return in full force. In fact, VX-770 withdrawal syndrome has been described in PwCF with the G551D mutation after cessation of Kalydeco™ treatment [337]. In addition, the currently available CFTR modulator treatments have not been able to fully halt disease progression. Long term follow-up studies of Kalydeco™—which has been on the market the longest, since 2012—have shown that lung function continues to decline, although at a slower pace compared to the control group [108]. Gene therapy thus holds potential to cure PwCF. Ever since the discovery of the *CFTR* gene in 1989, this has somewhat been the holy grail of causal CF therapies. Recently, the Cystic Fibrosis Foundation announced its “Path to a Cure”, which aims to find a curative treatment for all PwCF [338]. 

Early DNA targeting efforts focused on the delivery of a *WT*-*CFTR* cDNA to airway cells. Unfortunately, clinical trials with these gene addition therapies were not able to substantially improve lung function in PwCF (reviewed in [339,340,341]). This disappointment resulted in a shift away from gene therapy approaches for a while, but more recently, interest has been revived. Using optimized constructs and delivery methods, effective gene therapy has been shown in a number of CF models [342,343,344,345]. The most recent gene therapy clinical trial by Alton et al. demonstrated that liposome delivery of *CFTR* cDNA stabilized lung function in treated patients, whereas a further decline was observed in the placebo group [346]. As the clinical effect was considered modest, current efforts are focused on a F/HN-pseudotyped lentiviral vector to deliver *CFTR* cDNA to patients’ airways (BI 3720931, [344]). The first clinical trial for this viral vector-based gene addition therapy as well as several more developed by other groups are currently being planned [347,348,349]. 

Besides gene addition, gene editing, e.g., “rewriting” the mutation in the genome to the WT sequence, has gained a lot of traction over the last couple of years. The Nobel Prize for Chemistry in 2020 was awarded to Jennifer Doudna and Emmanuelle Charpentier for their discovery of the CRISPR/Cas9 system—the most versatile and commonly used gene editing technique [350,351]. It has been successfully employed in correcting *CFTR* mutations in translational cell models, such as rectal organoids [352,353,354] and airway epithelial progenitor (basal) cells [355,356]. We recently reviewed gene editing in CF and its potential for therapeutic approaches and model generation in detail. Therefore, we refer to [267,357] for an extensive discussion on the subject. 

## 4. Towards the Future: Personalizing Therapies for PwCF

In the previous section, we have provided an overview on the past and ongoing efforts in the development of novel causal therapies for CF. The majority of these efforts has focused on providing an effective therapy for F508del. This is not surprising, as such a therapy is able to treat ~85% of all PwCF. On top of that, the first market approval of a highly effective therapeutic strategy only became available as recent as 2019 (2020 in Europe) with the FDA and EMA approval of Trikafta™/Kaftrio™ for PwCF with one F508del allele. Previously, Orkambi™ and Symdeko™/Symkevi™ had already been approved for F508del homozygous PwCF, but with modest effects compared to Trikafta™/Kaftrio™. While the main focus for causal treatments has mostly been directed towards F508del, G551D became the first *CFTR* mutation for which a causal treatment (Kalydeco™) was market approved [61]. After additional clinical trials, treatment was extended for 9 more mutations both in the USA and Europe: R117H (c.350G>A; 3rd most common mutation in the United States), G1244E (c.3731G>A), G1349D (c.4046G>A), G178R (c.532G>A), G551S (c.1651G>A), S1251N (c.3752G>A), S1255P (c.3763T>C), S549N (c.1646G>A) and S549R (c.1645A>C) [63,358]. G970R (c.2908G>C) was also evaluated, but no improvements were observed for this mutation [63]. It was later shown that this particular mutant causes a previously undetected splicing defect rather than a gating defect, which precludes efficient rescue by CFTR potentiators [359]. Nevertheless, of importance was the fact that pre-clinical models, and more particularly, Fischer rat thyroid (FRT) cells overexpressing the respective mutations from a cDNA copy, were able to predict the outcomes for eight out of nine gating *CFTR* mutations tested [63,109]. These first label extensions were limited to *CFTR* mutations that were still (relatively) frequent in the sense that they allowed at least two PwCF per genotype to be included in the clinical study, and that their defects—at least to some extent—had been studied preclinically first. 

### 4.1. Theratyping & Expanding the Label for Existing Therapies

There are hundreds of *CFTR* mutations which are present only in a handful of PwCF and which have not or minimally been characterized. This poses two problems. First, as these mutations remain uncharacterized, how can the right therapeutic strategy be selected? On the one hand, the CFTR2 project aims to confirm and determine which of the >2100 mutations described to date in the *CFTR* gene are disease-causing or rather, polymorphisms or disease modifiers [360]. Full characterization via in-depth study of the folding, trafficking, function and stability of each mutation (a summary of these efforts is provided in [54]), would of course provide the best understanding of each mutation’s molecular defect, but might not be feasible to perform on thousands of mutations. Moreover, in order to be able to treat PwCF, it is more important to know whether particular mutations respond to certain treatments. Therefore, CFTR modulator therapy is now often tested directly on patient-derived material, a strategy called theratyping [361]. It allows us to (1) predict individual responsiveness to treatment (personalized medicine) and (2) gain insight into molecular mechanisms of rare mutations based on modulator responses [361]. One example of the use of theratyping is the ongoing European H2020 HIT-CF project (https://www.hitcf.org/ (accessed on 10 May 2022)), in which novel CFTR modulators are tested in human rectal organoids with rare genotypes and in vitro responders are selected for specific clinical trials, with hopes of bringing causal therapies to small, specific PwCF populations with rare mutations [362]. CFTR modulator responses are measured in the so-called forskolin induced swelling (FIS) assay [363]. Application of forskolin to these 3D, self-organizing organoid structures results in the rapid increase in organoid area, in the case that CFTR is functional. This process is dependent solely on CFTR, which is located at the apical membrane facing the organoid lumen and regulates paracellular water transport that causes the swelling of organoids. FIS responses have been shown to correlate well with clinical parameters such as the FEV_1_ (forced expiratory volume in one second, a measure for lung function) [364,365]. FIS can also be performed on organoids of airway origin, but due to the presence of other ion channels in the airways, the swelling is less CFTR specific and thus less straightforward to interpret [366]. Rectal organoids can be obtained via minimally invasive rectum biopsies and can be expanded rapidly and near infinitely, as such providing an advantage over other state of the art cell models, such as HBE (from explant lungs) or human nasal epithelial cells (HNE; from nasal brushings). The former are, when grown at air liquid interface (ALI), considered the golden standard for pre-clinical evaluation of CFTR function via short circuit current (I_SC_) measurements in Ussing chambers before moving towards clinical trials (for example [90]). Clancy and colleagues remarked that HBE analysis was missing for therapies that ultimately failed in clinical trials such as PTC124 and cavosonstat (*see Section 3.3.4. TRIDs & Section 3.2.3. Stabilizers* [361]). HBE, however, typically originate from end-stage disease tissue and are therefore not easily available for rarer genotypes [367]. HNE, on the other hand, do not have the above limitations and also have been shown to predict CFTR modulator responses [368], although the culturing is technically more challenging, requiring an important expansion step prior to testing [361]. The RARE project is an ongoing clinical trial which aims to collect intestinal and nasal cells of PwCF with rare mutations for future theratyping efforts (NCT03161808). Finally, also iPSC-derived airway models can be applied for theratyping [369]. 

The second question is: how can the efficacy of a treatment for rare genotypes be proven? The many rare to ultra-rare CF-causing mutations do not allow for classical clinical trial designs due to the small number of PwCF for each mutation. Alternative strategies are thus needed for label extension or novel therapies targeted towards the rare *CFTR* genotypes. For Kalydeco™, Symdeko™, and Trikafta™ label extension, the FDA therefore based itself on pre-clinical data from FRT cells overexpressing *CFTR* variants to guide its decision [62,64]. As a result, 97 mutations are now eligible for Kalydeco™ [65], 155 for Symdeko™ [370] and 178 for Trikafta™ [371]. In total, 184 *CFTR* mutations thus have modulator therapy approved (there is overlap between the mutations approved for each treatment), but it is important to note that not all the mutations listed are considered CF-causing according to the CFTR2 database. To date, label extension based on pre-clinical data has not yet been followed by the EMA.

### 4.2. What Is in the Pipeline for the Last 15% of PwCF without Causal Treatment?

Now that highly effective CFTR modulator therapy is available for the majority of PwCF, the focus has shifted towards the ~15% of PwCF who carry two alleles still without causal treatments, e.g., non-F508del, non-gating, non-residual function and non-rescuable by Trikafta™/Kaftrio™. While this group encompasses many, many rare and ultra-rare mutations, it is noticeable that of the top five most common *CFTR* mutations, only F508del (1st) and G551D (3rd) have an approved causal therapy to date. 

N1303K (4th) and I507del (c.1519-1521delATC; 11th) are two processing mutations with allele frequencies >0.5% that are not yet approved for modulator therapy [59,119,371,372]. In particular, N1303K has been investigated extensively [373]. In recent years, modest rescue was observed by modulator combinations in multiple cell models [114,119,138,374,375]. A case study reported a clinical benefit in one N1303K carrying PwCF with Trikafta™/Kaftrio™ treatment after 10 months, without a clear reduction in sweat chloride [375]. A clinical trial is currently ongoing to test its efficacy in 20 PwCF with the N1303K mutation (NCT03506061). R334W is a Class IV conductance mutation that is associated with some residual function and a milder form of CF. Nevertheless, it is currently not approved for modulator therapy as its in vitro rescue by Kalydeco™ was below the set threshold of >10% of WT in order to allow label extension [65,109]. Based on positive results in rectal organoids, a small clinical trial will test CFTR modulator therapy for this mutation (NCT04254705). 

PTC mutations G542X (2nd) and W1282X (5th) have become major targets for the development of CF causal strategies. Novel therapies are currently in pre-clinical and clinical development to tackle the specific defects associated with PTCs, i.e., NMD inhibition and TRIDs (*see Section 3.3.3. NMD Inhibition & Section 3.3.4. TRIDs*). As it is unlikely that NMD inhibition and TRIDs will fully be able to rescue nonsense CFTR, they will likely be combined with modulator therapy already on the market, such as Trikafta™/Kaftrio™. Pre-clinical studies so far suggest that TRIDs are also able to rescue rarer mutations such as Y122X (c.336T>A), R553X (c.1657C>T) and R1162X [268].

What remains are the “unrescuable” mutations. On the one hand, this comprises canonical splice mutations such as 621+1G->T and 1717-1G->A, the 7th and 8th most common *CFTR* mutations, respectively. In contrast to cryptic splice mutations with various degrees of alternative splicing, thereby retaining a proportion of correctly spliced mRNA, this fraction is almost non-existent with canonical splice mutations. Exons get alternatively spliced or skipped altogether, and the “blueprint” for the CFTR protein is gone [376]. In addition, in contrast to cryptic splice mutations, which are located further away from exon boundaries and therefore allow splice site modulation more easily, little progress has been made in repairing canonical splice mutations. As discussed above, splice-altering ASOs are expected to enter clinical trials soon [329,377], but for canonical splice mutations, it is more likely that precision gene editing will be needed to re-write the mutation to its WT sequence. To date, cryptic splice sites have been removed by knocking-out the mutation and surrounding sequence using CRISPR/Cas9 and Cas12a approaches [352,378]. These Cas-induced targeted, double stranded DNA breaks and subsequent DNA repair by non-homologous end-joining lead to effective knockout (KO) of the cryptic splice site by indel (insertion-deletion) formation, which is possible due to their location sufficiently deep inside an intron. Since canonical splice mutations reside right at exon-intron boundaries, KO by indel formation is not precise enough and hence not an option. Base or prime editing, both CRISPR-derived gene editing technologies (reviewed in [267,357]), allow to make precise, targeted edits in the genome, and are more likely able to correct these mutations. In that regard, successful adenine base editing has already been already reported for correcting nonsense mutations in *CFTR* [354,379]. In addition, frameshift mutations caused by small indels might be rescued by base or prime editing approaches in the future. Collectively, these CRISPR/Cas technologies have, to date, mainly focused on the repair of F508del, PTC mutations and 3849+10kbC>T [353,354,379], and can be further expanded to study the repair of most *CFTR* mutations (reviewed in [267]). Larger deletions, however, such as the 21kb deletion mutation CFTRdele2,3 cannot be targeted by this type of strategy and will require other corrective means such as gene addition (reviewed in [339]), super-exon insertion (see [356,380] for examples, although neither super-exon designs include the region deleted in CFTRdele2,3), stimulation of alternative chloride channels or ENaC inhibition (reviewed in [381,382]). While gene editing has reached the clinic for several diseases (for an overview of its clinical development we refer to [383,384]), this is not yet the case for CF. 

## 5. Conclusions

In this review, we have aimed to cover the diversity of *CFTR* mutations and the different therapeutic approaches that have been, and are currently being, developed to rescue all mutations. The driving force behind the dedicated CF community is to find a treatment and eventually a cure for all PwCF. Tremendous progress has been made since the first description of the *CFTR* gene in 1989, which has led to the approval of four CFTR modulator therapies in the last decade. With these, PwCF carrying one F508del allele, or one of the additional 183 approved mutations can be treated. The focus has now shifted to providing causal treatment for the remaining 15% of PwCF. For PwCF carrying PTC mutations, a novel TRID is currently being clinically evaluated. NMD inhibition strategies have taken flight, although they have not yet reached the clinical stage yet. Rapid advances in the field of gene editing might, for the first time, provide a way to treat the underlying gene defect in mutations previously considered unrescuable. New gene and mRNA replacement clinical trials are planned that could provide a mutation agnostic approach for all PwCF. Novel CFTR modulators are being developed to improve CFTR rescue and reduce side effects, as well as proteostasis modulators, stabilizers and amplifiers. All in all, exciting times remain ahead for the causal treatment of CF.

## Figures and Tables

**Figure 1 cells-11-01868-f001:**
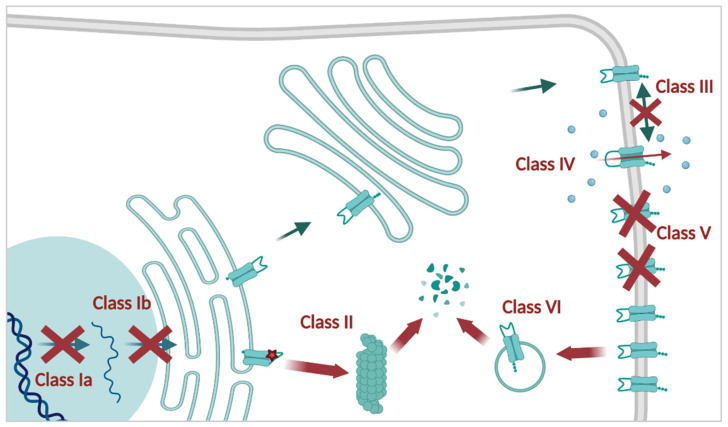
CFTR mutations can cause different defects and are grouped accordingly. Green arrows: normal CFTR biogenesis and function. Red arrows/crosses: defects caused by the mentioned mutation classes.

**Figure 2 cells-11-01868-f002:**
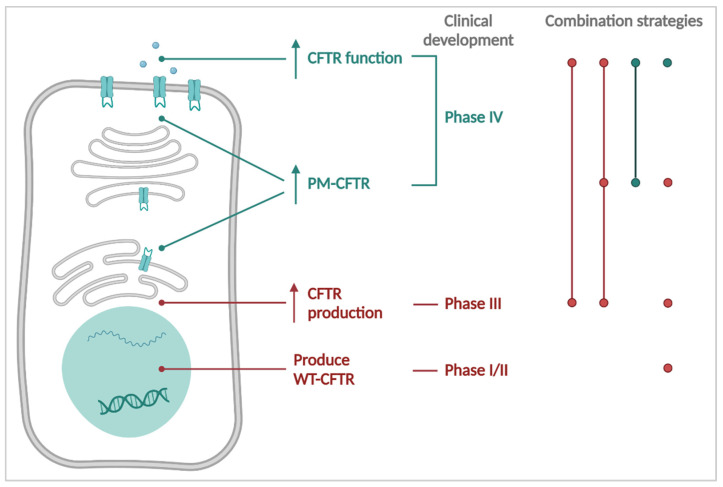
Global overview of therapeutic strategies for causal treatment of CF and their current progress in clinical development. On the right, strategies are represented by dots, and lines connecting them indicate combination therapies, either market-approved (green), or in clinical development (red). WT: wild-type. PM: plasma membrane.

**Figure 3 cells-11-01868-f003:**
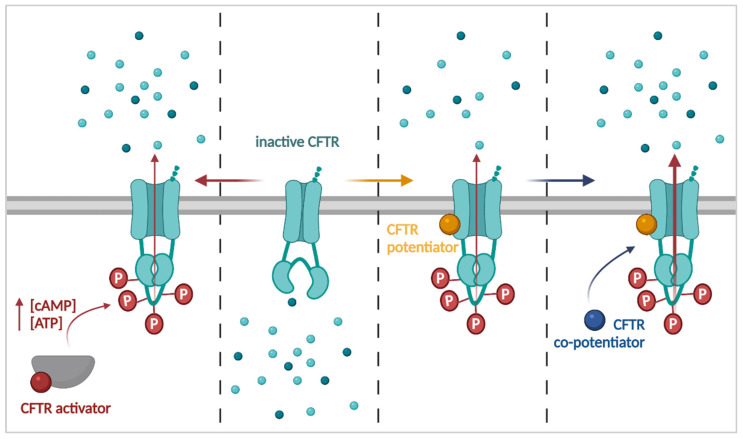
Strategies to improve CFTR function. Left: Activators (red) improve CFTR function by altering its normal regulation, for example by promoting phosphorylation of CFTR or via ATP analogues which lock CFTR in an open state. Right: CFTR potentiators (orange) interact directly with CFTR to promote function, in a phosphorylation-dependent, but ATP-independent manner. Co-potentiators (blue) also interact with CFTR, to further stimulate function but only in the presence of CFTR potentiators.

**Figure 4 cells-11-01868-f004:**
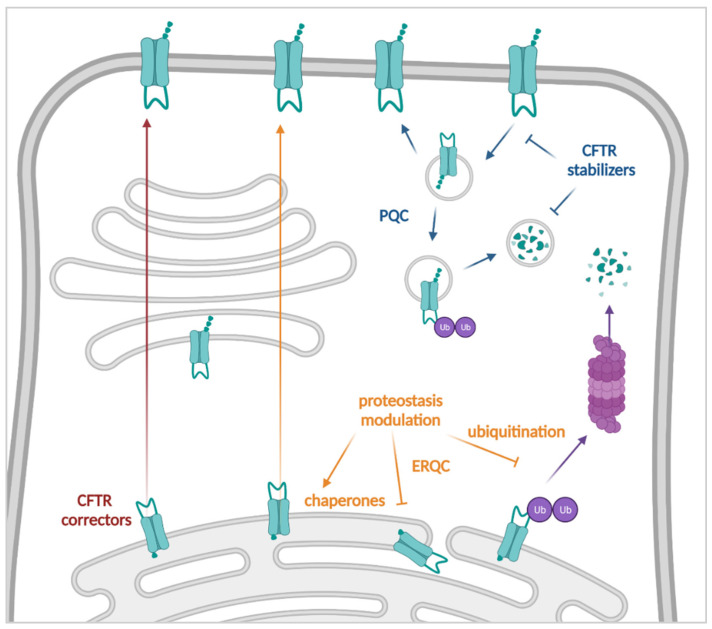
Strategies to improve the amount of CFTR channels at the plasma membrane (PM). CFTR correctors (red) interact directly with CFTR to improve its folding and subsequent trafficking. Targeting CFTR interactors through proteostasis modulation (orange) allows mutant CFTR to escape early degradation and traffic to the PM. Stabilizers (blue) prolong the duration of CFTR’s residence at the PM by slowing down its turn-over and peripheral degradation. ERQC: Endoplasmic Reticulum Quality Control; PQC: Peripheral Quality Control.

**Figure 5 cells-11-01868-f005:**
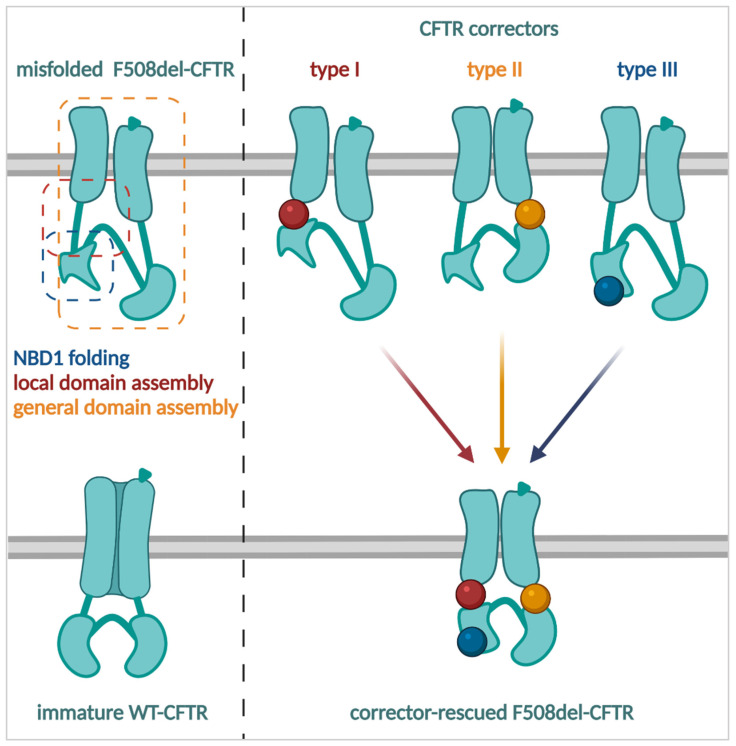
Different types of CFTR correctors. Left: The F508del-CFTR protein is misfolded and does not achieve the native conformation of wild-type (WT)-CFTR. Schematic overview of folding defects associated with the F508del mutation. Right: Type I, II, and III correctors tackle different folding defects, which alone or in combination (partially) rescue F508del to a more WT-like state.

**Figure 6 cells-11-01868-f006:**
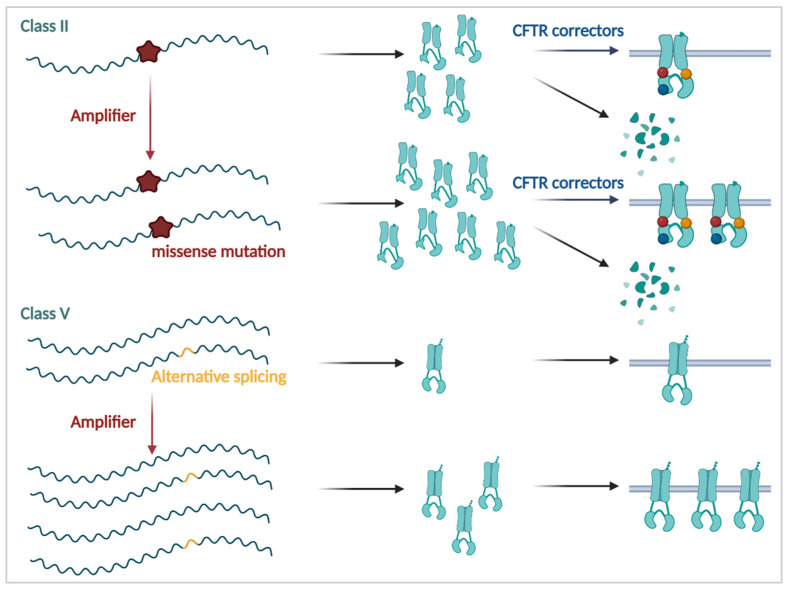
Amplifiers stabilize the *CFTR* mRNA and thereby increase translation, enlarging the pool of immature CFTR protein. Amplifiers (red) are of particular interest for the treatment of Class II and V mutations. Top: Class II mutations are misfolded and only a fraction reaches the plasma membrane, even in the presence of CFTR correctors (blue). Amplifiers provide more immature protein that can subsequently be rescued with CFTR modulators. Bottom: Class V mutations introduce cryptic splice sites, resulting in a mix of normal and alternatively spliced (yellow) *CFTR* mRNA. The fraction of normal mRNA gives rise to wild-type CFTR, but the number of channels is severely reduced. Enhancing the number of correct mRNA transcripts results in more CFTR protein at the PM.

**Figure 7 cells-11-01868-f007:**
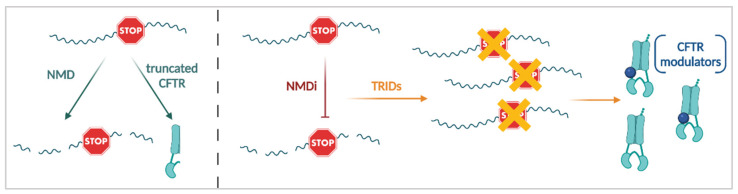
*CFTR* with premature termination codon (PTC) mutations: defects and therapeutic strategies. Left: PTC *CFTR* mutations present two specific defects: the mRNA is often degraded via nonsense mediated mRNA decay (NMD), and the resulting protein is truncated. Right: To overcome these challenges, NMD inhibitors (NMDi) and Translational Readthrough Inducing Drugs (TRIDs) are investigated. As TRID treatment does not necessarily result in incorporation of the native amino-acid, these treatments can further be complemented with CFTR modulators.

**Figure 8 cells-11-01868-f008:**
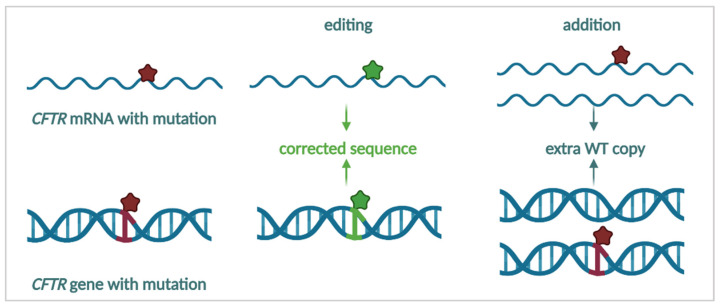
Strategies to produce wild-type (WT) CFTR by introducing the correct gene or mRNA sequence. The principles for the strategies for mRNA and DNA are identical, either the existing, mutated version is edited to the WT sequence, or a new correct exogenous copy is introduced into cells. Red star: sequence with mutation; green star: corrected sequence.

**Table 1 cells-11-01868-t001:** Overview of *CFTR* mutations in this review, their frequency and approved treatments.

Name	cDNA Name	Allele Frequency (%)	CFTR2 Alleles	Approved Treatment
CFTRdele2,3	c.54-5940_273 + 10250del21kb	0.3	417	NA
P67L	c.200C>T	0.2	239	K/S/T
G85E	c.254G>A	0.4	616	T
E92K	c.274G>A	<0.1	49	S/T
R117H *	c.350G>A	1.3	1854	K/S/T
Y122X	c.366T>A	0.1	88	NA
621+1G>T	c.489+1G>T	0.9	1323	NA
G178R	c.532G>A	0.1	87	K/S/T
R334W	c.1000C>T	0.3	429	NA
W496X	c.1487G>A	<0.1	3	NA
I507del	c.1519-1521delACT	0.5	651	NA
F508del	c.1521-1523delCTT	69.7	99061	O/S/T
1717-1G>A	c.1585-1G>A	0.9	1216	NA
G542X	c.1624G>T	2.5	3610	NA
S549R	c.1645A>C	0.1	93	K/S/T
S549N	c.1646G>A	0.1	203	K/S/T
G551S	c.1651G>A	<0.1	19	K/S/T
G551D	c.1652G>A	2.1	2986	K/S/T
R553X	c.1657C>T	0.9	1323	NA
2789+5G>A	c.2657+5G>A	0.7	1027	K/S
G970R	c.2908G>C	<0.1	12	NA
R1162X	c.3484C>T	0.5	651	NA
I1234V	c.3700A>G	<0.1	33	NA
3849+10kbC>T	c.3718-2477C>T	0.8	1158	K/S
G1244E	c.3731G>A	0.1	106	K/S/T
S1251N	c.3752G>A	0.1	120	K/S/T
S1255P	c.3763T>C	<0.1	10	K/S/T
W1282X	c.3846G>A	1.2	1726	NA
N1303K	c.3909C>G	1.6	2246	NA
Q1313X	c.3937C>T	<0.1	30	NA
G1349D	c.4046G>A	<0.1	22	K/S/T
Q1412X	c.4234C>T	<0.1	4	NA

NA: no treatment available; K: Kalydeco™; O: Orkambi™; S: Symdeko™; T: Trikafta™. * mutation with varying clinical significance. Frequency/allele numbers: www.CFTR2.org (accessed on 10 May 2022); Approved treatments: www.kalydeco.com (accessed on 10 May 2022)/www.orkambi.com (accessed on 10 May 2022)/www.symdeko.com (accessed on 10 May 2022)/www.trikafta.com (accessed on 10 May 2022).

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
