# Peer review of "One Size Does Not Fit All: The Past, Present and Future of Cystic Fibrosis Causal Therapies"

_cells, 2022, doi:10.3390/cells11121868_

Round 1
Reviewer 1 Report
The manuscript submitted by Ensinck and Carlon is indeed a very nice piece of work, detailing the history of CF therapies aimed to correct CF basic defect.
The review provides a comprehensive overview of all the different types of CFTR-modulating compounds and drugs that have been used to investigate CFTR biology and to restore CFTR processing/function. The manuscript also beautifully summarizes the potential role of gene- and mRNA-based therapies for those who carry unrescuable CFTR variants and thus still lack a cure.
The references are very accurate and complete. The only very minor point is the schematic drawing in Figure 2, which is a little bit confusing. Is there a way to simplify it? Or maybe to split them into two panels, one devoted to the MoA and the other to the possible combination?
Reviewer 2 Report
In this manuscript, Ensinck & Carlon provide an excellent overview of targeted therapies available or in development to correct the basic defect causing cystic fibrosis. There are however a few minor aspects that should be corrected or added - in such a comprehensive review with almost 400 references (!), one would expect that “all” relevant literature is cited.
1. When referring to PwCF for which modulators are not available yet, authors should prefer a percentage around 15% - if we consider the European Registry (and add other populations in which F508del prevalence is even lower), 10% is not that representative.
2. In line 39, replace “anion” by “ion” – as cation imbalance (e.g. sodium) is also an hallmark of CF.
3. When referring to the mutation classes, it would be better to use the classification proposed by DeBoeck & Amaral (ref. 42), as it clearly distinguishes mutations that are “unrescuable” from those in which chemical/pharmacological rescue can be pursued.
4. For the class IV mutations (line 164-166), it would be advisable to use another example rather R117H – this variant is only disease-causing when combined with the IVS8 5T polymorphism.
5. Figure 2 is misleading regarding the “site of action” for correctors – as they supposedly act on ER-retained protein. Please adjust accordingly.
6. When referring to mutations that completely abolish CFTR function (either nonsense, trafficking, gating), please avoid the word “minimal”. It was coined by the pharmaceutical function Vertex, but it doesn’t reflect the reality – there is no function at all (which is different from a minimal one, which suggests that there is some function remaining).
7. In Figure 4, please add the lysosome as a major site for degradation for endocytosed CFTR that does not recycle back to the PM.
8. When referring to mechanisms for stabilizing CFTR at the PM (lines 642-651), the authors may consider adding recent works that report the regulation of spleen tyrosine kinase SYK activity (PMID: 31974654), the activation of the cAMP sensor EPAC1 (PMID: 27206858 and 32573649) or the inhibition of the protease calpain (PMID: 31324722) as mechanisms for CFTR PM stabilization.
9. When discussing the use of NMD inhibitors, a comment should be added on the toxicity of molecules such as SMG1i.
10. When discussing read-through approaches, studies using engineered tRNAs should be mentioned (PMID: 30778053 and doi 10.1016/j.omtn.2022.04.033).
11. When referring to gene therapy approach, the results of the non-viral trials performed by the UK CF Gene Therapy Consortium need to be mentioned (PMID: 26149841) – even if the team decide not to pursue further this approach and to shift to a viral approach instead.
Reviewer 3 Report
In the review by Ensinck et al, the authors examined the past and ongoing therapeutical approaches to rescue CFTR protein. The authors did a great job giving an extensive overview of the small molecule combinations used as a strategies to rescue different CFTR mutations. This is a remarkably well-written review and is interesting. Congratulations to this outstanding study!
Author Response
We thank reviewer 3 for reviewing our manuscript and their kind words with respect to our manuscript.